# Knowledge of health benefits, availability, accessibility and consumption of indigenous foods by urban adults in the Cape Metropole, South Africa

Tasneem Johnson[1], Mthokozisi Zuma[1], Laurencia Govender [2], Nelene Koen[1], Xikombiso Mbhenyane [1]*

1 Division of Human Nutrition, Faculty of Medicine and Health Sciences, Stellenbosch University, Tygerberg Campus, Cape Town, South Africa, 2 Department of Dietetics and Human Nutrition, School of Agricultural, Earth and Environmental Sciences, University of KwaZulu-Natal, Pietermaritzburg Campus, Pietermaritzburg, South Africa

* xgm@sun.ac.za

## Abstract

### Background

Indigenous foods are effective in addressing malnutrition and food insecurity and contain phytochemicals which have been reported to prevent non-communicable diseases. Understanding the role that indigenous foods play can add to our understanding of how to promote these foods within urban communities of South Africa.

### Aim

To describe the knowledge of health benefit, availability, accessibility, perceptions and consumption regarding South African indigenous foods.

### Methods

A cross-sectional study design was used. Data was collected using a managerial survey (n=5), observational checklist (n=27), consumer survey (n=43) and household survey (n=344). Adults between 18 and 80 years residing in the Cape Metropole participated.

### Results

Most of the household's population had a low-medium dietary diversity (65.8%). Perceptions regarding indigenous foods were mostly favourable, with economic and nutritional benefits and the quality reported as being key aspects. Indigenous foods were reported to be most available from supermarkets (69.0%) as opposed to grocery stores (19%) or food stalls (8.0%). Samp, maize meal and pumpkin observed at food establishments were seen to be the most accessible indigenous food items.

**Data availability statement:** Data is available from the corresponding author and the Institution upon request. This is because of the requirement for publication consent and Material Transfer Agreement and Data Transfer Agreement Health Research Ethics Committee requirements and processes that need to be completed. The data underlying the results presented in the study are available from Prof XG Mbhenyane of the Division of Human Nutrition, Stellenbosch University, South Africa. www.sun.ac.za. The Institutional contact for the Health Ethics Research Committee: Ms Elvira Rohland Principal Administrator: Health Research Ethics Tel: 021 938 9677 E-mail: elr@sun.ac.za http://www.sun.ac.za/english/faculty/healthsciences/rdsd/Pages/Ethics/Contact-us0715-8777.aspx.

**Funding:** The research was made possible with grants from the South African National Research Foundation (Grant Identifications: 98954 and 118563). The funders had no role in study design, data collection and analysis, decision to publish, or preparation of the manuscript.

**Competing interests:** The authors have declared no competing interest exist.

Many households (57.6%) fell below the monthly income required to ensure food security. Only 58.8% and 47.2% of household participants reported knowing indigenous foods and their health benefits, respectively.

## Conclusion

The study population demonstrated limited knowledge regarding the definition and health benefits of IFs. Within the Cape Metropole, IFs were perceived as largely unavailable and inaccessible across food establishments. Additionally, there was a common misconception equating IFs with exotic foods. Despite this confusion, a significant proportion of participants expressed a willingness to purchase IFs if they were more readily accessible.

## Introduction

Numerous studies have shown the positive role of indigenous foods (IFs) on food and nutrition security in rural areas [1–3], the same wealth of research is not available for urban environments where a similar diet-related disease profile is persisting. Additionally, food markets are sensitive to climate extremes [4], and these extremes are becoming more common in South Africa [5,6]. The well-adapted nature and health benefits of IF crops make them a highly promising solution to addressing chronic health conditions and food and nutrition security concerns [4,7]. The consumption of indigenous fruits and vegetables is inversely related to disease risks such as cardiovascular disease, cancer, coronary heart disease, infection, hypertension, type 2 diabetes mellitus, obesity and overall mortality risk due to their impressive phytochemical content and antioxidant properties [7,8]. Indigenous foods' potential in preventing and managing these conditions are of great importance. In addition, research shows that the health benefits associated with IFs are known to community members, and this may act as a positive driver in promoting the use of IFs [9]. For these reasons, IF usage is particularly relevant to the burden of disease context seen in South Africa today [7].

A study by Kucich and Wicht [10] showed that South African indigenous fruits contain a high level of antioxidants. The fruits assessed were some of the most easily available and accessible indigenous fruits, including wild plum, crossberry, waterberry, sour fig, *num-num*, *bietou* and tortoise berry. Fruits such as the wild olive, colpoon and Christmas berry were also seen to have high antioxidant profiles, but these fruits are used for medicinal purposes only due to their bitter taste [10].

Akinola et al. [11] define a host of nutritional benefits associated with IF crops that can play a major role in addressing hunger, malnutrition and other nutrition-related diseases. South African indigenous plant foods have a higher protein and other nutrients content than the most popular exotic food crops. Many indigenous vegetables also have a higher nutritional value than well-known exotic vegetables such as tomato and cabbage [12]. Indigenous foods are also a nutrient-dense form of food

and offer a healthy alternative to many of the food items available in Westernized markets. Akinola et al. [11] noted that most South African IF crops can offer vitamins, macro and microelements in amounts that exceed the World Health Organization's recommendations for a healthy diet for adults and children [11]. They are thus an ideal source of macro and micronutrients that can provide nutrition to many.

The study by Kesa and Mbhenyane [13] aimed to assess the knowledge, perception, and consumption of IFs among residents of Gauteng, South Africa, one of the country's most urbanized regions. Using a quantitative cross-sectional survey of 746 participants, the researchers found that while certain IFs like grain sorghum and marula were relatively well-known, actual consumption was low and largely seasonal. Consumption patterns varied significantly by cultural group, with Black participants consuming more IFs than other groups. Despite limited availability, participants expressed strong interest in increasing IF consumption, and negative perceptions of IFs were minimal across all cultural groups.

Studies have also identified accessibility and availability as major factors affecting IF consumption, while unfamiliarity and poor perceptions regarding these foods were seen to have a significant negative impact on their consumption [1,14]. Thus, the earmarking of knowledge, perceptions, accessibility, and availability of IFs as areas of interest regarding IF consumption allows for a comprehensive assessment of the issues at hand and ensures the development of more robust conclusions and solutions [5,15,1].

Indigenous foods can enable every South African to achieve their "five a day" recommendation for fruits and vegetables with less reliance on economic aspects, making the concept of food for all people ever more possible [1,2]. Their adaptations to the South African climate are fundamental to their myriad benefits, which extend into sustainable agriculture and trade [16]. The efforts of researchers in understanding the status of IFs in South Africa provides insight into channels for learning and improvement in the Cape Metropole [11,17–23]. However, the status of IFs is still an understudied area in the Cape Metropole area and Western Cape.

## Methods

### Study design

An observational study design that used a cross-sectional data collection procedure was employed [24,25]. This study aimed to obtain information of the status of IFs amongst adults between 18 and 80 years residing in the Cape Metropole. Variables that expressed the availability, accessibility, perceptions, knowledge about health benefits and consumption of IFs were observed.

### Setting and population

The research took place in the Cape Metropole of the Western Cape of South Africa. Of the 278 municipalities in South Africa, the Cape Metropole has the second-largest population (3,740,026 individuals, 42.4% coloured, 38.6% African, 15.7% white, 1.4% India/Asian and 1.9% other). Despite its economic standing, the Cape Metropole still has a 23.9% unemployment rate, with 35.7% of its households living below the poverty line [26].

Adults between the ages of 18 and 80 years residing in the suburbs of the Cape Metropole make up 74% of the population, according to Statistics South Africa [27]. Only participants living in formal dwellings and residing in urban environments were selected. Participants were from the various ethnic groups of South Africa (Asian, African, coloured and white) and the various socioeconomic groups (low, medium and high socioeconomic groups).

### Study period

The study data was collected from 17/02/2020, once ethical clearance was obtained, and recruitment ended 16/02/2021. However, the entire project was initiated in 02/02/2019 and concluded 12/12/2022 including reporting.

## Sampling

### Sampling approach

Multi-stage sampling was employed for data collection. The eight districts of the Cape Metropole were included as a sampling frame. Purposive sampling was employed to select suburbs that provided a cosmopolitan diversity of the Cape Metropole. These suburbs were identified with the assistance of a statistician, using an online map and diversity statistics of the Cape Metropole. Only seven of the eight districts were sampled since one district did not include any suburbs with the required diversity spread. After purposive sampling of districts, 17 suburbs of the Cape Metropole were selected. These 17 suburbs included an extra suburb that was used for the piloting of the study.

For the market survey (MS), purposive sampling was employed to select one supermarket, grocery store and food stall from each suburb. One manager or store owner from each food establishment was approached to participate in the study. Fieldworkers conducted the observational survey (OC) within each of the selected establishments, where permission was granted to do so. For the consumer survey (CS), fieldworkers were stationed in the fresh produce aisle of the previously selected establishments, or alongside the establishment in the case of food stalls, after receiving permission from store managers or owners. Fieldworkers approached candidates shopping in the fresh produce aisle of supermarkets or grocery stores or those approaching food stalls. Consumers who were willing to participate and met the inclusion criteria were asked to provide informed consent. Thereafter, fieldworkers conducted a short interview with the consumer.

Household survey used systematic sampling for the household and convenience sampling for one adult willing participant. Systematic sampling was done to select streets to recruit household participants. Suburb maps were used to identify the streets within the 17 different suburbs. The street closest to the Central Business District (CBD) was identified as the first street to be sampled for data collection. Successive streets that fell within the suburb boundary line were then be sampled after the first street. Fieldworkers recruited participants from houses on either side of the street by ringing the doorbell, knocking on the door, or approaching residents who were standing outside of their homes. Initially, a systematic sampling of houses was selected, where every second household was sampled. However, due to the COVID-19 pandemic and other general challenges regarding approaching candidates' homes, every house in the street was sampled to allow for adequate sample sizes within each suburb. Convenience sampling was used to select the household informant for the survey interview who was an adult and responsible for food distribution.

Convenience sampling was employed to recruit focus group discussion (FGD) participants. At least one FGD was planned per suburb with a minimum of five and a maximum of twelve participants to allow for a large enough sample. As part of the household survey, participants were asked if they would be willing to participate in an FGD.

See Fig 1 below with the visualisation of the sampling process.

(Source: Authors)

### Sample size

For the MS, each establishment's manager or store owner was selected to participate. The calculated sample for manager participants was thus: 1 manager per establishment x 3 establishments per suburb x 17 establishments = 51 manager participants. One observation was conducted per establishment. Therefore, the calculated observation sample size was: 1 observation per establishment x 3 establishments per suburb x 17 suburbs = 51 establishments. For the CS, convenience sampling was used, with the research team proposing that a minimum of two consumers be sampled per establishment. Thus, the calculated sample size was: 2 consumers per establishment x 3 establishments x 17 suburbs = 102 consumer participants. The household sample size was calculated using Slovin's formula [28] at 95% confidence interval and p-value of 0.05. The final sample comprised 440 households. From this, 10% (n = 44) were selected to participate in focus group discussions (FGDs), representing the designated sample size for qualitative inquiry.

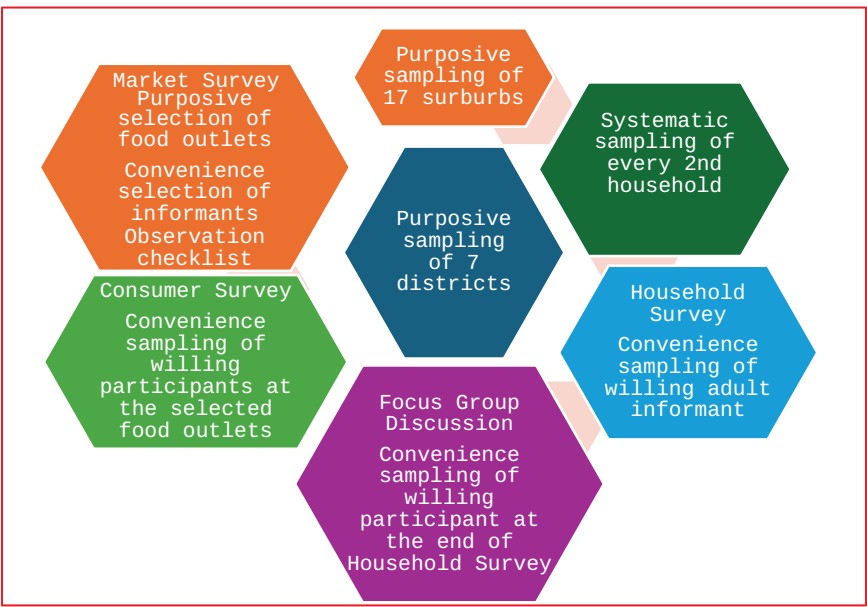

**Fig 1. Multi-stage sampling Process.** (Source: Authors).

### Inclusion and Exclusion criteria for household participants

Adult males and females between the ages of 18 and 80 years who resided in the Cape Metropole and were involved in purchasing or preparing food for the household were included in the study. The broad age range allowed for a larger sampling pool while including the elderly population, who are important sources of knowledge regarding IFs [6,29]. Individuals were excluded if they had any mental conditions such as Alzheimer's or Dementia that would affect their ability to give accurate responses. Fieldworkers ascertained this prior to beginning the questionnaire by asking the participants or their family members in instances where participants showed signs of impaired cognition. Where an individual did not provide consent to participate, they were also excluded from participation.

## Measurements and techniques

The literature available with regards to IFs locally [11,30–32], nationally [11,33–37] and internationally [38,39] was reviewed. Findings were used to advise tool development for the MS, OC, CS, HHS and FGD. A list of 47 IFs was generated according to the literature [31,33,37,40] and included in the MS, OC and HHS to allow for triangulation of results. Tools were translated from English to both Afrikaans and IsiXhosa to allow for data collection amongst all groups that spoke the most spoken languages in the Cape Metropole. Internal testing, piloting and subsequent adaptation of each data collection tool took place in a two-step process. Tools were first tested by fieldworkers amongst themselves. This was followed by pilot testing in the pilot suburb selected by the research team.

### Market survey

The market survey was conducted using store managers, observation by researcher and field workers and interview of consumers.

**Managerial survey.** The frequency of procurement per month, demand per month, source of procurement (locally or imported) and quantity wasted per month were identified as measurements that would provide insight into how available

food establishments made IFs and how well these IFs were managed within the establishment. Also, after considering research from abroad [7,15,39,41] and within South Africa [6,30,34,40,42], supermarkets, grocery stores and food stalls were identified as the main suppliers of fresh produce of which IFs would form part of. Supermarkets and grocery stores were considered as formal food establishments while food stall were designated informal in this study. A managerial survey was developed from this information.

**Observational checklist.** The same IFs used in the managerial survey were used in the development of the OC. From the literature, the quantity, quality and price per kilogram of IFs were identified as key areas for assessment to determine the availability and accessibility of IFs at food establishments [11,15,31,33]. From this information, a checklist was developed, which included all the 47 IFs as well as space to record the quantity, quality and price of each IF that was available. A reputable guide to assessing the quality of fresh produce was identified [43] and advised the development of the tool.

**Consumer survey.** A survey including open-ended and closed-ended questions was developed after referring to current literature [9,10,29,30,34,42,44] regarding the relationship between consumers and IFs. This consumer survey evaluated the availability and accessibility of IFs in the marketplace as well as consumers' perceptions regarding IFs. A previously developed questionnaire that included both open-ended and closed-ended questions was used to develop the CS tool [19].

## Household survey

After consulting the literature and developing the market survey, a household survey was developed. The household survey included separate sections of questions to assess sociodemographic information, availability, accessibility, knowledge of health benefits, perceptions and consumption of IFs.

The consumption section made use of a 24-hour recall determining household dietary intake, as well as a checklist of foods and food preparation methods to assess the intake of IFs by participants over the previous month. The remainder of the sections made use of open- and closed-ended questions. The participant's knowledge surrounding IFs was determined using a semi-structured questionnaire which was adapted from an existing questionnaire [42].

## Focus group discussions

The focus group discussion included five open-ended questions that were developed from the literature [9,10,29,30,34,42,44], and in accordance with the MS, OC, CS and HHS. These questions related to IFs availability, accessibility, knowledge of preparation methods, perceptions and knowledge health benefits. A focus group discussion guide, which included all these questions as well as brief instructions on how to implement was developed.

## Training and standardization

Seven fieldworkers were appointed, with the principal investigator acting as the eighth fieldworker. The research team arranged a one-day training session in which fieldworkers were orientated towards the study's background, aims and objectives. During this training, fieldworkers were also introduced to the data collection tools and allowed to practise on each other. Fieldworkers were selected to allow for interviewing in isiXhosa [2], Afrikaans [6] and English [8], and all fieldworkers had at least a matric qualification, with one having graduated university and two having postgraduate qualifications.

## Covid-19 risk mitigation plan

The COVID-19 pandemic posed a challenge to data collection as potential participants were hesitant to take part in the study due to concerns regarding contracting COVID. A COVID-19 Risk Mitigation Plan was developed prior to data collection and implemented throughout data collection.

## Pilot study

A pilot study was done to review the data collection process and included all the intended questionnaires (MS, OC, CS and HHS). One of the preselected suburbs was chosen due to its central location regarding all fieldworkers. Study techniques were adapted as required following the results of the pilot study. None of the pilot study participants were considered for participation in the main study.

## Ethical and legal aspects

### Permissions

Ethical approval was obtained from the Health Research Ethics Committee of Stellenbosch University (HREC No: S19/05/091). Approval to conduct research within the Cape Metropole was also obtained from the City of Cape Town. The study design and procedures were also in accordance with the World Medical Association Declaration of Helsinki and the Constitution of the Republic of South Africa [45,46].

Written informed consent to participate was obtained from each household, managerial and consumer survey participant. Each participant received a copy of the consent form for future reference. Permission was obtained from the manager of each supermarket/grocery store/food stall prior to the commencement of data collection, which included permission to collect data outside the stores. Consumer informed consent was also obtained from shoppers. Household participants were asked to provide written or verbal informed consent prior to participation. All data collected was used only for ethically approved purposes.

### Data analysis

Dietary diversity scoring was done using the 24-hour recall. After capturing, data were summarized into a dietary diversity score (DDS) using a 12-point modification of an existing DDS tool [47]. Where the DDS score was below three, dietary diversity was low, where it was four to five, it was medium, and where it was above five, it was considered high. The frequencies and percentages for each category were generated using the Statistical Package for the Social Sciences IBM SPSS Statistics Version 29, and data were analysed according to these values. The descriptive statistics were analysed for each variable and across variables to identify trends.

Where it was deemed necessary by the research team, variables were controlled with regard to age, gender, ethnicity and economic status to eliminate bias in the results. Ethnicity was controlled only to eliminate the influence that cultural eating habits would have on the study outcomes.

Data were analysed by generating descriptive statistics that demonstrated means, standard deviations, frequencies and percentages.

Thematic analysis was used to analyse qualitative data [48]. In instances where open-ended question responses were captured, similar responses were grouped together, and themes were identified according to these groupings. These themes were interpreted by the principal investigator.

## Results

### Sociodemographic information

For the HHS, a total of 344 households were sampled with a response rate of 86%. More than half of the participants were aged between 25 and 45 years old (58.1%, n = 200). There were more male participants (n = 203) 59.0% than females (n = 137) 39.8%. There was a good cultural distribution typical of the Cape Metropole. Almost half of the sample (42.4%, n = 146,) came from five suburbs within the Northern Suburbs district, with over one-fifth of the sample coming from four suburbs within the Southern Suburbs district (21%, n = 72, 21%).

For the market survey, a total of 5 managers (9.8% response rate) and 43 consumers (42% response rate) were sampled, and a total of 27 observations (100% response rate) were made. Three grocery store owners/managers completed a survey, as well as one supermarket and one food stall owner/manager. The three store types planned were covered in the sample. Amongst the observational sites, the majority (64.3%, n = 18) were done with grocery store owners/managers. Most consumers were surveyed at supermarkets (74.4%, n = 32) (Table 1).

## Indigenous food availability information

With regard to food establishments, of the 47 items included on the OC, only 10 were observed. Consumer participants were only asked about fresh produce IF items. The white-fleshed sweet potato was seen in food establishments by all consumer participants.

According to household participant responses, butternut (95.0%, n = 324), maize meal (92.7%, n = 316), and pumpkin (89.7%, n = 305) were the most available IFs where household participants bought their fresh produce.

## Indigenous food accessibility information

From in-store observations, samp was the most affordable indigenous food at 0.68 USD per kilogram, followed by maize meal at 0.81 USD per kilogram and pumpkin at 1.06 USD per kilogram. The quality of fresh produce was seen to be medium to high amongst supermarkets and grocery stores, with a high quality observed in IFs sold at food stalls.

Most consumer participants never purchased indigenous fruits at all (79.07%, n = 34). More than half of consumer participants (55.81%, n = 24) reported that they had seen some of the IFs listed where they purchased their fresh produce. Most consumer participants (70%, n = 28) stated that the items sold in the fresh produce section where they usually purchased their fresh produce were of good quality and always looked fresh.

Amongst households, 20.9% (n = 72) had an income of between 529.03 and 1587.05 USD and 30.8% (n = 106) households had an income of less than 264.52 USD. Nearly half (48.6%, n = 167) reported that less than half of their household's monthly income went towards purchasing food. Between 105.81 and 158.66 USD was spent on food by (25.6%,

**Table 1.  Demographic spread of household and market survey participants.**

| Household participants | | | | Market survey participants | | | |
|---|---|---|---|---|---|---|---|
| Characteristics of household participants | Demographic variables | n | % | Market survey tool | Type of establishment | n | % |
| Age (n = 344) | 18-25 | 32 | 9.3 | Managerial survey (n = 5) | Supermarket | 1 | 20 |
| | 25-29 | 73 | 21.2 | | Grocery store | 3 | 60 |
| | 30-45 | 127 | 36.9 | | Food stall | 1 | 20 |
| | 46-59 | 46 | 13.4 | Observational checklist (n = 27) | Supermarket | 5 | 18.5 |
| | 60-80 | 40 | 11.6 | | Grocery store | 18 | 66.7 |
| | Do not want to answer | 26 | 7.6 | | Food stall | 4 | 14.8 |
| Gender (n = 344) | Female | 137 | 39.8 | Consumer survey (n = 43) | Supermarket | 32 | 74.4 |
| | Male | 203 | 59.0 | | Grocery store | 8 | 18.6 |
| | Other | 4 | 1.2 | | Food stall | 3 | 7.0 |
| Cultural group (n = 344) | African | 115 | 33.4 | | | | |
| | Coloured | 144 | 41.9 | | | | |
| | Indian | 14 | 4.1 | | | | |
| | White | 54 | 15.7 | | | | |
| | Other | 17 | 4.9 | | | | |

n = 88) of the households. More than half of the households (52.7%, n = 177) purchased fruit weekly, (22.0%, n = 72) purchased fruit monthly, and (16.4%, n = 55) purchased fruit every other week. Sixty-six participants (19.7%) reported purchasing indigenous vegetables when in season.

### Perceptions regarding indigenous foods

Perceptions were assessed in the market survey via the consumer survey. The consumer survey employed open-ended questions to provide information on consumers perceptions regarding IFs.

Consumer participants were asked if any in-store factors influenced whether they purchased IFs. Twelve consumer participants cited food quality while nine cited knowledge surrounding IFs, and eight stated that no factors influenced whether they purchased IFs. This shows that quality and knowledge are areas of interest with regards to perceptions regarding indigenous foods.

### Indigenous food knowledge information

Household participants were asked if they knew the difference between indigenous and non-indigenous food and if they knew the health benefits of IFs. These open-ended questions were used to determine household participants knowledge regarding IFs as well as their knowledge regarding IF health benefits.

Of the total study sample, 58.1% (n = 200) and 46.5% (n = 160) reported knowing what IFs were and what their benefits were, respectively. Thus, about half of the participants were knowledgeable regarding IFs.

### Indigenous food consumption information

Household participants were asked to complete a 24-hour recall in which they listed which foods they had eaten the day before and how much of each food they had eaten. Consumption was assessed using a 12-point DDS score [44] which categorized household participants into low, medium and high dietary diversity categories. See Table 2 below.

### Indigenous food perceptions information

Household participants' perceptions regarding IFs were assessed by examining their responses to various statements relating to IFs and exotic foods. See Table 3 below.

## Discussion

The discussion is arranged according to the themes: IF knowledge on health benefits, availability, accessibility, consumption, perceptions and limitations.

### Indigenous food knowledge and health benefits

There was an overall lack of knowledge regarding IFs amongst the study population. While more than half of the household participants reported knowing what an IF was, less than half of these participants gave an accurate definition. This is

**Table 2. A summary of household participants' Dietary Diversity Scores (DDS).**

| DDS* range | n | % |
|---|---|---|
| Low DDS | 68 | 21.9 |
| Medium DDS | 136 | 43.9 |
| High DDS | 106 | 34.2 |
| Totals | 310 | 100 |

*DDS: Dietary Diversity Score.

in keeping with the results of studies done by Chakona and Shackelton [30] and Akinola *et al*. [11] that showed a persisting loss of IF knowledge from one generation to the next, threatening the long-term sustainability of food systems in South Africa.

Many of the participants that answered "yes" when asked if they knew the difference between IFs and non-IFs gave answers that related to the origins, availability and accessibility of these foods. Some also responded with answers relating to the sustainability and nutritional value of these foods. In these instances, participants were correct in their responses. Amongst those who did not know what IFs were, responses were varied and showed significant misconceptions. Some participants reported that they thought the difference between IFs and non-IFs lied in how these foods were grown and the quality of these foods. For example, some participants thought that IFs were, in fact, organic or non-genetically modified (non-GMO) foods. Other participants thought that IFs made one's stomach run or that they were, in fact, exotic foods. These misconceptions regarding IFs are the result of the loss of IF knowledge that occurs both with urbanisation and Westernisation, which can be seen within South Africa [49].

When compared to those who reported knowing the difference between an IF and non-IF, fewer participants reported knowing the benefits of IFs. When asked if they knew the health benefits of IFs, less than half of the participants said yes. Of those who reported knowing what the benefits of IFs were, more than 80% provided answers that were related to the benefits of IFs. Of the participants who said they knew the benefits, the majority provided answers that related to availability, economic, health, cultural and environmental benefits, which were correct [49]. Some participants cited factors such as IFs being organic, which was not correct. This demonstrates that a larger proportion of participants were not aware of the benefits of IFs, and some were significantly mistaken with regards to IF health benefits.

Together, these findings demonstrate that there is some understanding amongst the study sample with regards to what IFs are and the benefits of IFs, but also that a large proportion of the study sample was not aware of what an IF was or the benefits of IFs. This is in keeping with a study by Akinola *et al*. [11] that found that IF knowledge amongst the South African population was restricted due to the relocation of communities under the Apartheid regime. This relocation resulted in communities having limited access to IFs and disrupted the continuation of IF cultivation as well as the knowledge associated with these foods. The differences seen in knowledge regarding what IFs are and the health benefits of IFs may stem from the differences in education seen across ethnic groups in Cape Town, the result of Apartheid legislation [50]. However, knowledge regarding IFs was not significantly influenced by cultural group, which demonstrates an overarching lack of knowledge amongst different cultural populations of the Cape Metropole. This finding contrasts with studies that have shown that cultural factors are drivers in IF knowledge accumulation and IF usage [51–54]. However, this study was urban-based and not representative of South Africa or cultural groups in rural settings. Since the age distribution of the study population is lower than what is estimated for the general population of the Cape Metropole [26], the lack of knowledge observed may be due to fewer elderly individuals being included. Another reason for this may be South Africa's rapid rate of urbanization, which has produced a food system reliant on commercial agriculture and supermarket provision of food [55]. This has led to poorer diets amongst the population, but it also holds the key to boosting IF provision in urban areas. Due to the well-established and dependable supermarket chains operating in South Africa, a viable entry-point for

**Table 3. A summary of household participant responses regarding indigenous foods.**

| Statement | Responses | Total | % |
|---|---|---|---|
| **Indigenous foods and exotic foods are the same** | Strongly disagree/disagree | 200 | 58.1 |
| **Indigenous foods are better than exotic foods** | Not sure | 131 | 38.1 |
| **Exotic foods are better than indigenous foods** | Strongly disagree/disagree | 146 | 42.4 |
| **I prefer to buy indigenous foods** | Strongly agree/agree | 180 | 52.3 |
| **I prefer to buy exotic foods** | Not sure/ disagree | 186 | 54.1 |

stimulating IF provision within the main food system already exists to take advantage of. By increasing the provision of IFs through these channels, IF usage can be addressed on a large scale. Understanding the role of indigenous foods by the population and linking them to the health benefits such as prevention of non-communicable diseases, may help promotion of their consumption.

These results emphasize the general disconnectedness of communities with IFs and speak to the findings of studies done by Mbhenyane [36] and Weinberger [41] that defined IFs as undervalued foods. These findings also concur with the results of a study done by Akinola et al. [11], which reported that one of the main sociocultural benefits of IFs was to connect communities with their cultural "roots". This is related to the fact that IFs have historically acted as cultural resources that form part of various cultures and ways of life of communities [52]. Since the study population is from a community (South Africans), the majority of whom have been previously restricted from IFs through relocation from their traditional environments, their cultural connection with these foods has diminished, and this has resulted in their knowledge regarding these foods being decreased.

Knowledge regarding IFs is key in translating intention to use these foods into practice. The findings of this study that demonstrate a lack of knowledge amongst the population are concerning since knowledge regarding IFs plays a big role in its acceptance and usage within communities [30,33,40]. With less knowledge regarding these foods, promoting the use of IFs as an economical and sustainable means to address food insecurity in urban areas will be a much bigger challenge.

## Indigenous food availability

The study highlighted various IFs as being available across the Cape Metropole and identified the food establishments which were most likely to make these food items available to the community.

One manager response highlighted butternut and maize meal as being products that were in high demand, procured regularly, sourced locally and as having a high enough demand to result in zero wastage. This finding was confirmed through observation, with maize meal, samp and beans, white-fleshed sweet potato and samp being the most observed IFs across suburbs and at different types of food establishments. From household participant responses, butternut, maize meal, and pumpkin were the most reported IFs available in stores, followed by samp and samp and beans.

This demonstrates that butternut, maize meal, samp and beans, samp and white-fleshed sweet potato are the most widely available IFs. These findings show that the supply of IFs in the Cape Metropole is not in keeping with the abundance of IFs available in the country, as the South African list for underutilized foods cites green leafy vegetables as being the dominant food typically found in South Africa [23,56]. Yet green leafy vegetables do not feature with regards to availability in this study. The IFs reported in this study are starch based and do not reflect the full spectrum of indigenous underutilised crops. It is noted that the Cape metropole is an urban municipality comparable to the other seven metropolitan municipalities. Most of the studies to date have been done in rural settings with more access to the indigenous varieties [36]. The IFs available may reflect the demand in the area, which could be influenced by cultural eating habits and food preferences [19]. Availability may also be affected by the agricultural context of the Cape Metropole since some, but not all South African IFs grow well in the climate of the Cape Metropole. Some IFs may also not be part of the food production and supply chains, and this affects their production quantities and the steadiness of their supply in markets. Another consideration is the fact that IFs are perishable, and their availability is influenced by their transport and storage. Within the context of the Cape Metropole, as well as their specific food establishment, food establishment owners must consider perishability before deciding which fresh food items to procure. Thus, the individual circumstances of food establishments are an important consideration with regards to the status of IFs in the Cape Metropole. Transport and storage procedures regarding IFs also need to be investigated for all food establishment types to identify whether conditions are optimized for IFs. Loubie Rusch of the Sustainability Institute at Stellenbosch University has been exploring the forgotten and neglected wild foods of the Cape since 2010. She has foraged and observed them in nature, has grown, cooked and bottled them and has shared about them. As time passed, her wild food journey evolved into finding ways for local

indigenous foods to contribute to recalibrating the social, economic and ecological imbalances and contribute to climate change impacts.

On observation, IFs were the most available through supermarkets, with grocery stores displaying a much smaller availability of IFs and food stalls a very restricted availability of IFs. Supermarket one, Supermarket two and Supermarket three were highlighted by household participants as stores where IFs were generally available. Amongst consumer and household participants, supermarkets were the most popular type of food establishment at which participants purchased fresh produce, with more than half of household participants reporting to shop for fresh produce at supermarket one. These results indicate that supermarket one is a store well-known for having IFs available to purchase and that supermarkets, in general, have better availability of IFs. This finding identifies food markets and specific stores that can potentially be investigated to build on the understanding of how IFs can be made more available in mainstream food markets. These findings are also, however, contrary to findings of a study that assessed Sub-Saharan Africa's availability of IFs in which it was shown that informal food markets were the main source of IFs [11]. The reasons for the difference seen between South Africa and Sub-Saharan Africa, in general, are important to investigate to highlight opportunities that will allow South Africa to improve its informal food market supply of IFs. In addition, it would be beneficial to understand the lack of the informal market trade in green leafy vegetables as fresh and dried when compared to the starches observed in this study. These findings also show that IFs are not readily or broadly available within and across food markets in the urban setting, which agrees with the findings of studies done by Mabhaudi [33] and Hadebe [56]. These studies indicate that although certain IFs may be well used in other countries within Sub-Sharan Africa, their utilisation within urban South Africa is significantly low due to unavailability, which has led to urban South Africa's IF usage being limited to very few food items.

Home food gardens also influence the access that households have to fresh produce [20,57–60]. Most households reported not having a home food garden, with only one-quarter indicating they maintained one. Those with gardens were asked to list the foods they cultivated. Among these, more than half reported growing only exotic produce, while the remainder cultivated a mix of exotic and IFs. These findings relate to the results of a study done by Bvenura and Afolayan (352) that showed that IFs were mostly foraged with little effort amongst individuals to cultivate them for use as food. Similarly, Misselhorn [10] also showed that in many cases in South Africa, social grants provide enough income to address food insecurity while simultaneously discouraging home food production. Missing the opportunity to cultivate IFs adds to the IF gap, and this gap can easily be filled by the exotic foods which are much more available through South African food markets. These findings highlight the much higher availability of exotic foods in South Africa, which is in keeping with findings of a study done by Akinola et al. [11]. The findings regarding food gardens are particularly interesting since IFs are known to grow better in the South African climate and soils than exotic foods and would thus be more productive if cultivated in home food gardens [11,61]. The preference for exotic foods indicates that the convenience and benefits of IFs are undervalued within the study sample, which is in keeping with the study findings of Chivenge [7], Mbhenyane [36] and Weinberger [41]. Studies have shown home food gardens to be an effective means by which to address food insecurity at a household level [10,62]. With less land available for large scale agriculture in urban areas, increasing IF availability through food gardens is ideal, but it also faces the obstacle of ensuring adequate food production within a limited area. Further research must investigate successful food garden initiatives in urban areas and identify how these operations function and how their processes can be promoted and implemented across urban communities.

## Indigenous food accessibility

Food accessibility refers to the price, quality and preference for available foods. The accessibility of IFs was considered by examining their cost and quality across various food establishments in the Cape Metropole. Amongst the sample, a limited number of IFs were adequately accessible across markets, and supermarkets generally offered the most affordable forms of IFs. Interestingly, not all IFs that were the most available were seen to be the most affordable within the study.

According to the price per kilogram, the most affordable IFs in the Cape Metropole were maize meal, samp and pumpkin, with larger quantities of maize meal and samp being more cost-effective.

This finding agrees with the fact that maize meal and samp were amongst the most widely available IFs in the Cape Metropole because their high demand would allow these IFs to be affordable [11]. Maize meal is also a staple food in South Africa which is part of the mainstream food system [36]. Maize is grown as a cash crop, intended to be sold in food markets in large quantities or to be exported to other countries. This allows for maize meal to have much more competitive pricing than other IFs, which are not produced on the same scale. White-fleshed sweet potato was one of the most accessible and consumed IFs [63]. However, white-fleshed sweet potato's accessibility was much lower than less available IFs like samp. This finding leads us to ask why other IFs that are much less available than white- fleshed sweet potato are not less accessible than white-fleshed sweet potato? This may have to do with the fact that white-fleshed sweet potato is one IF that is seen to be imported more than other Ifs [64]. With regards to quality, another reason for this may be that the robustness of canned samp and packaged maize meal means that these foods maintain their quality better during transportation and storage, which improves their accessibility [63]. Another reason for this may be that the demand for maize meal and samp is much higher amongst consumers, which allows for these IFs to be sold at a more competitive price than food items which are less in demand [65].

We speculate that reasons for butternut, maize meal, samp and beans, samp and white-fleshed sweet potato as the most widely available IFs is due to their perishability and that they are available. The same is applicable for green leafy vegetables not featuring with regards to availability in this study. Otherwise, the discussion is very sound with scientific tone.

General household income and food spending behaviour provided information on how economically accessible IFs were to households. More than half of the sample were the main breadwinner in their own home and responsible for purchasing groceries, with more than one-third of households earning a monthly income of between 264.52 and 1587.05 USD. This implies that a large proportion of the study sample were employed, which adds to their ability to access available foods, including IFs [11,32]. One-third of the sample reported that the grocery-purchasing responsibility was shared in their household, with a smaller proportion of participants reporting that their partners were responsible for purchasing groceries.

Most participants specified answers relating to knowledge and traditional roles when asked why a certain individual within the household was given the responsibility of purchasing groceries, while some participants cited economic status and family dynamics as being important factors. This shows that traditional factors still play a strong role in food-acquiring behaviour, which is in keeping with studies done by Mavengahama [66] and Yang [61]. Very few households relied on the elderly, children or individuals outside of the household with regards to procuring groceries. Thus, in general, the participants in this study were economically capable and at low risk with regards to accessing the IFs available in stores. Almost 80% of the sample did not receive social grants, which is in keeping with a study done by Chakona and Shackelton [30], which found that social grants were least common amongst urban dwellers when compared to rural and peri-urban dwellers. In urban areas, job availability is higher than in rural areas. This higher availability of jobs means that the urban dwellers in the sample are more likely to have an income that would assist in accessing food regularly. For this reason, the influence of social grants on the accessibility of households to IFs was not significant in this study.

With regards to purchasing behaviour, household participants generally purchased fruit and vegetables weekly, but more than half of participants said that they did not know if their household purchased indigenous fruits or vegetables, with a smaller proportion reporting that they only bought indigenous fruits and vegetables when they were in season. With regards to purchasing behaviour, consumer participants generally purchased fresh produce once a week but only purchased indigenous vegetables once a month or less than once a month, with most never purchasing indigenous fruits at all. This finding is interesting given the fact that more than half of consumer participants reported having seen these IFs at the store where they usually purchased their fresh produce, and 70% stating that the fresh produce available was of good

quality and always looked fresh. Thus, factors other than availability and quality have an influence on whether individuals buy IFs [11,30,61]. However, only one-third of consumer participants stated that IFs were available when they purchased their fresh produce, and almost half reported sometimes having to go to another store to find the IF they were looking for.

Access plays a significant role in shaping the availability of IFs, particularly in urban retail settings. As noted, many IFs are highly perishable, which can discourage supermarkets and food stalls from stocking them due to concerns about storage and wastage. It is also important to highlight that IFs are predominantly produced by small-scale and subsistence farmers, who often face significant barriers in accessing formal markets. Historical regulations and procurement practices continue to favor commercial producers, effectively excluding smallholder farmers from mainstream supply chains. We hope that increased consumer demand and awareness will help shift these practices, encouraging more inclusive sourcing and improved availability of IFs across food establishments.

These findings imply that IFs are not frequently accessed by adults in the Cape Metropole, but also that IFs may not be steadily accessible within the main markets, which is in keeping with the results of studies done by Akinola *et al*. [11] and Hadebe [56]. However, just under half of the participants also stated that they never had to go to another store to find the IF item they were seeking. This variation in responses may be related to the different types of IFs that individuals shopped for. It may also add to the implication that IFs were not frequently accessed by individuals due to their own food preferences.

In general, this purchasing behaviour shows a preference for exotic fruits and vegetables, which concurs with findings of research done by Akinola *et al*. [11] and Bvenura and Afolayan [34]. It also shows that indigenous starches were much more commonly purchased than indigenous fruits and vegetables amongst the study population.

## Indigenous food consumption

Overall consumption of IFs follows on from the lower knowledge, availability and accessibility of IFs in comparison to exotic foods.

Almost two-thirds of the study sample had low to medium dietary diversity, and just over one-third of the sample had high dietary diversity. This is similar to findings of studies done on South African populations, which showed average dietary diversity to be low-medium across three different cities in South Africa [16,63,66–68]. This indicates that in this sample, dietary intake is limited in variety and similar to the general population of South Africa. In addition, in the previous month, participants were seen to have eaten significantly more exotic foods than IFs, with the most eaten exotic foods being eaten by almost twice as many participants when compared to the most eaten IFs. Amongst the total study participants, cereals, spices, condiments and beverages as well as eggs, meat and vegetables were the most eaten food items, and this demonstrated that diets were generally low in vegetables, fruit, legumes, nuts and seeds and fish. This agrees with a study done by Chakona and Shackelton [30] who reported low intake of fruits and vegetables. A vast array of IFs fit into the less frequently eaten food groups. Thus, the low dietary diversity seen in this sample will significantly affect individuals' intake of IFs.

These findings agree with Akinola *et al*. [11], who showed that IF intake was low in South Africa, with rural communities consuming a wider variety and larger quantity of IFs. The urban-rural differences in consumption are influenced by the availability of the IFs. Research by Gakobo and Jere [69] also found that IF consumption was decreasing in some sub-Saharan countries, and this may be the case amongst this study sample. Future research also needs to examine the role ethnicity and culture play in IF consumption in urban areas.

A similar 2023 study by Kesa et al. [13], which examined knowledge, perception, and consumption of IF) in Gauteng, revealed that while awareness of certain Ifs, such as grain sorghum, marula, pearl millet, *amadumbe*, and cowpea, was relatively high, actual consumption remained low, with grain sorghum being the most consumed by less than one-fifth. Consumption was largely seasonal and varied across racial groups, with Black South Africans consuming more IFs than others. Importantly, participants expressed a strong willingness to increase IF intake, and negative perceptions were not a

major barrier; instead, limited availability was the primary challenge. These findings align closely with those from the Cape Metropole, where IFs are also underutilized despite positive perceptions and a willingness to consume them. The Gauteng study reinforces the need for targeted interventions, particularly in education and market access, to bridge the gap between intention and practice. For the Cape Metropole, this suggests that improving availability, enhancing nutritional education, and culturally contextualizing IFs could significantly increase their integration into urban diets and contribute to addressing pressing health challenges such as malnutrition, non-communicable diseases, and food insecurity.

**Indigenous food perceptions**

Perceptions regarding IFs shape the way that individuals interact with these foods and is also a gatekeeper to establishing individuals' knowledge regarding these foods. A study by Gakobo and Jere [69] showed that, while consumers had high intentions to consume IFs, this intention did not translate into practice. A similar trend was seen amongst the study population in this regard.

Most participants did not think that IFs and exotic foods were the same. Regardless, almost half of the participants also said that they were not sure if IFs or exotic foods were better. At the same time, most participants said that they did not prefer to buy exotic foods. Quality and knowledge were cited by many participants with regards to their outlook on IFs. More specifically, quality of IFs, knowledge surrounding IFs, availability, cost and health benefits of IFs were all cited as factors that influenced whether participants bought IFs. Many participants believed that IFs had superior health benefits and quality, moreover they were cheaper to buy than other foods. These perceptions are contrary to studies that have shown that individuals felt that making use of IFs was associated with being poor or that IFs were less appealing foods [11,30]. This finding is promising since it shows positive perceptions amongst this study sample with regards to IFs. This outcome also showed that many participants thought that a lack of knowledge surrounding IFs was one of the main factors influencing their usage of IFs. This is in keeping with a study done by Akinola *et al*. [11] that cited a lack of knowledge and a lack of market access as the main factors affecting the usage of IFs in South Africa. The study by Akinola *et al*. [11] also found that negative perceptions regarding IFs were a major challenge to IF usage. However, this was not seen amongst this study population.

Culture plays a meaningful role in IF perceptions as well. IF food systems are known to represent more than just nutritional resources. They include cultural aspects that influence individual perceptions regarding these foods and whether they are used by individuals and communities [49,52,69]. Numerous studies have shown that cultural heritage and cultural knowledge affect IF food usage in India [70], the United States of America (USA) [52], and South Africa [49]. In this study, culture, represented by ethnicity, was not seen to play a significant role in individuals' perceptions regarding IFs. However, this does not mean that culture does not play a role in this sample. A study by Shanks *et al*. [52] showed that dietary patterns, food choices and nutrition-related diseases were shaped by ecology, economics, politics as well as culture. Thus, myriad factors come into play regarding food perceptions. In this study, knowledge regarding IFs was seen to vary according to cultural group [70]. In this way, cultural grouping could influence IF perceptions and use. The study by Kesa et al. [71], in Gauteng, found that most participants lacked knowledge about where to obtain IFs, with vendors being the most common source and supermarkets, schools, and workplaces offering limited access. This aligns with findings from the Cape Metropole, where IFs were also found to be restricted in availability and accessibility, particularly when compared to exotic foods. However, both studies revealed that despite these limitations, perceptions of IFs remained positive, and there was a strong willingness among urban populations to increase their consumption. The Cape study further highlighted that even poorer households and informal food establishments relied more on exotic foods, despite the potential economic advantages of IFs. These parallels suggest that urban South African communities share similar barriers to IF integration, primarily driven by limited market presence and low public awareness. Both studies underscore the need for targeted interventions, particularly in nutrition and health education, market reform, and culturally relevant promotion strategies. to transform IFs from undervalued resources into integral components of sustainable urban diets.

Studies done in the USA and South Africa showed that a complexity of cultural diversity exists within and across cultural groups and that this cultural diversity influences dietary patterns [38,51,54]. Browne *et al*. [38] emphasized that effectively addressing health disparities in the USA requires more focus on cultural and psychosocial processes as well as the social, historical and economic context of communities. It can be assumed that this approach is also relevant to South Africa, where there exists a rich cultural diversity. In the study, the investigation may not have adequately assessed a wide enough range of the cultural factors that influence food perceptions, and this may be why a link between culture and IFs was not seen.

## Limitations of study

Many of the grocery stores and food stalls in the Cape Metropole are owned or run by foreign nationals whose first language was not one of the languages used in the study. Future research should allow for these individuals to participate in surveys by providing translators for the most spoken languages spoken by these individuals. This will not only provide better insight into IFs in the Cape Metropole but also IFs in Africa since some of these individuals cultivate IFs from their own countries in South Africa to sell in South African food markets.

The study did not capture sociodemographic information of the individuals who participated in the market survey. Thus, the age, gender, cultural grouping and economic status of managers and consumers cannot be compared to the household participants or provide insight into the differences seen between the managerial participant and consumer responses in comparison to household participants. Future research would need to include these sociodemographic parameters to gain a more comprehensive view of the status of IFs in the Cape Metropole.

The sampling pool was limited. Future studies should address this by increasing the sample size significantly and sampling individuals from all suburbs of the Cape Metropole to provide a better representation of residents and eradicate sample bias. Some suburbs were also excluded because additional permissions were required to perform research in these areas (for example, in the instance where suburbs were also naval bases). Future research should ensure that permissions to sample these areas are obtained.

In addition, had this study investigated why individuals who used IFs did so, we would have gained greater insight into the factors promoting IF usage in urban areas. Future research should include this type of investigation. Furthermore, the IFs observed were mainly starchy and not vegetables, thus findings cannot be compared with those done in rural areas where there is variety of IFs.

While numerous studies done in Africa, South Africa and abroad found that culture played a meaningful role in IF knowledge, the study did not find any significant link. However, the study included only a limited number of measures describing cultural groupings. Thus, we could not thoroughly investigate ethnicity's effect on IF use or assume that culture did not influence IF knowledge in the study population.

## Conclusion and recommendations

This study was able to establish the status of IFs in the Cape Metropole with regards to their availability, accessibility and consumption by individuals living in urban environments, as well as the knowledge and perceptions of these individuals regarding IFs and their health benefits.

### Knowledge

Overall knowledge of IFs was limited, with considerable confusion regarding both their definition and associated health benefits. Despite low consumption levels, perceptions of IFs were generally favourable. Participants identified knowledge and the quality of IFs as key factors influencing their usage. The status of IFs in the Cape Metropole remains poor, underscoring the need for targeted promotion across food markets and households. Furthermore, economic, educational, and

cultural factors, as well as the role of food markets, emerged as critical areas for intervention to enhance the uptake and integration of IFs.

Beyond economic considerations, the study revealed a significant lack of knowledge about IFs among adults in the Cape Metropole. Given that adults are typically responsible for educating minors, it is reasonable to infer that children in this urban area also lack awareness of IFs. This perpetuates a cycle of limited IF knowledge that is unlikely to be broken without targeted education efforts directed at adults and the broader community. These findings prompt critical questions: Why is IF knowledge so inadequate, and how has this gap affected individuals' ability to translate their willingness to incorporate IFs into actual usage? Compared to studies conducted in other sub-Saharan African countries, the study population appeared to lag in IF awareness. However, it is important to note that this sample reflects urban South Africa rather than the national population. The disparity may be attributed to differences in economic structures, agricultural systems, food market dynamics, or cultural practices. Further research is needed to explore these factors and identify successful strategies from across sub-Saharan Africa that could be adapted to elevate the status of IFs within South Africa.

### Accessibility

A limited variety of IFs were available in local markets, with accessibility varying significantly across different types of food establishments and IF categories. Consumption of IFs was notably low and substantially lower than that of exotic foods. Household dietary diversity was classified as medium. Supermarkets emerged as the primary source of IF provision in the Cape Metropole, while grocery stores and informal food stalls lagged considerably in their offerings

### Availability

In the Cape Metropole, IFs were found to be restricted in both availability and accessibility, particularly when compared to exotic foods. The variety of IFs available was limited, and even households with lower income levels, as well as informal food establishments, tended to rely more heavily on exotic foods. This trend persists despite the potential economic advantages of IFs, which are generally easier to cultivate and maintain agriculturally. These findings suggest a missed opportunity to leverage IFs as a cost-effective and culturally relevant component of local diets

### Consumption

The preference for exotic foods among the study population was evident, with the majority opting to purchase and consume these over Indigenous Foods (IFs). However, certain Ifs, such as maize meal, butternut, samp, samp and beans, pumpkin, and white-fleshed sweet potato, emerged as exceptions. These items were found to be relatively available, accessible, and culturally accepted within this urban population, indicating potential entry points for broader IF promotion and integration into urban diets.

### Perception

Interestingly, the limited availability and accessibility of IFs in the Cape Metropole have not resulted in negative perceptions among the population. On the contrary, a substantial proportion of individuals expressed positive attitudes and a willingness to increase their use of IFs. While IFs are generally available, albeit in lower variety and quantity compared to exotic foods, they remain accessible enough for community members to incorporate into their diets. This highlights the need for targeted interventions aimed at converting this willingness into actionable behaviour, thereby promoting the integration of IFs into everyday food practices.

### Recommendations

In addition to providing a cultural context for the promotion of IFs, education emerged as a key area for intervention. The study identified nutritional and health education as essential pathways for disseminating information about IFs within

communities. This need is underscored by the population's limited understanding of IFs, particularly regarding their definition and health benefits. By leveraging educational initiatives to promote IFs, their role in addressing major health challenges in South Africa, such as malnutrition, non-communicable diseases, and food insecurity, can be amplified and better appreciated. This approach has the potential to elevate IFs from undervalued to respected components of urban diets, encouraging their production, usage, and integration into daily life.

Culture, as a multifaceted and dynamic aspect of society, cannot be quantified simplistically. These findings suggest the need for deeper exploration of the relationship between culture, IF knowledge, and IF usage in future research. This is particularly relevant in the Cape Metropole and broader South Africa, where multicultural urban communities are the norm. By identifying cultural factors that influence IF awareness and consumption, and aligning IF promotion efforts with cultural relevance, IFs can be meaningfully contextualized within communities. Such an approach not only reinforces the historical and future significance of IFs but also highlights their potential contribution to sustainability across South Africa.

## Acknowledgments

The authors would like to acknowledge all participants from the Cape Metropole and the six research assistants.

## Author contributions

**Conceptualization:** Tasneem Johnson, Laurencia Govender, Nelene Koen, Xikombiso Mbhenyane.

**Data curation:** Tasneem Johnson, Mthokozisi Zuma.

**Formal analysis:** Tasneem Johnson, Mthokozisi Zuma, Nelene Koen, Xikombiso Mbhenyane.

**Funding acquisition:** Xikombiso Mbhenyane.

**Investigation:** Tasneem Johnson, Mthokozisi Zuma, Laurencia Govender, Nelene Koen, Xikombiso Mbhenyane.

**Methodology:** Tasneem Johnson, Mthokozisi Zuma, Laurencia Govender, Nelene Koen, Xikombiso Mbhenyane.

**Project administration:** Tasneem Johnson, Xikombiso Mbhenyane.

**Resources:** Tasneem Johnson, Xikombiso Mbhenyane.

**Software:** Mthokozisi Zuma.

**Supervision:** Laurencia Govender, Nelene Koen, Xikombiso Mbhenyane.

**Validation:** Tasneem Johnson, Mthokozisi Zuma, Nelene Koen, Xikombiso Mbhenyane.

**Writing – original draft:** Tasneem Johnson, Mthokozisi Zuma, Laurencia Govender, Nelene Koen, Xikombiso Mbhenyane.

**Writing – review & editing:** Tasneem Johnson, Mthokozisi Zuma, Laurencia Govender, Nelene Koen, Xikombiso Mbhenyane.

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
