## [Decision Letter · Decision Letter 0]

31 Jul 2025

Dear Dr. Mbhenyane,

Thank you for submitting your manuscript to PLOS ONE. After careful consideration, we feel that it has merit but does not fully meet PLOS ONE’s publication criteria as it currently stands. Therefore, we invite you to submit a revised version of the manuscript that addresses the points raised during the review process.

We look forward to receiving your revised manuscript.

Kind regards,

Edwin Hlangwani, PhD

Academic Editor

PLOS ONE

Journal Requirements:

4. We note you have included a table to which you do not refer in the text of your manuscript. Please ensure that you refer to Table 3 and 4 in your text; if accepted, production will need this reference to link the reader to the Table.

Reviewers' comments:

Reviewer's Responses to Questions

**Comments to the Author**

1. Is the manuscript technically sound, and do the data support the conclusions?

Reviewer #1: Yes

Reviewer #2: Yes

2. Has the statistical analysis been performed appropriately and rigorously?

Reviewer #1: Yes

Reviewer #2: I Don't Know

3. Have the authors made all data underlying the findings in their manuscript fully available?

Reviewer #1: Yes

Reviewer #2: Yes

4. Is the manuscript presented in an intelligible fashion and written in standard English?

Reviewer #1: Yes

Reviewer #2: Yes

Reviewer #1: Knowledge of Health Benefits, Availability, Accessibility, and Consumption of

Indigenous Foods by Urban Adults in the Cape Metropole, South Africa

The manuscript is well written and sounds very interesting. However, the researchers may need to consider some of the following comments and suggestions to improve the paper.

Some aspects of the paper are unclear, such as the difference between formal and informal supermarkets among the surveyed supermarkets. I think the type of supermarket may influence the food found, either exotic or indigenous. This is because various food products have different storage and transportation requirements. As a result, most informal supermarkets, particularly those owned by individuals in communities, lack the proper infrastructure to store certain food products, particularly perishable and semi-perishable products, which may lead to a poor supply of such products, regardless of whether they are IFs. Another critical aspect was the distinction between income levels and the type of food items purchased among the participants.

In most cases, affordability plays a crucial role in influencing buying patterns among communities; hence, the buying power in a given community also plays a part in determining the type of shops or supermarkets you could find. This is because low-income earners are most likely to buy key food products necessary to sustain them throughout the month, instead of purchasing healthy food products. Therefore, understanding the buying patterns based on income levels could help us understand why specific households prefer certain food products over others, particularly indigenous products over exotic ones. Furthermore, the study does not clearly explain the nutritional benefits of the IFs compared to the exotic ones readily available or accessible in the area or supermarkets using literature. Without the link between the IFs and their alternative foods offering similar benefits, it may be hard to say the participants prefer certain products over others, particularly if such products are not even related or used in place of each other, such as eating apples instead of guava. I think the products need to be comparable or offer the same benefits for one to say they prefer one over the other. Also, a link between the available IFs and their role in addressing malnutrition and non-communicable diseases, including their health benefits, would help show the health risks that could arise from the communities' eating habits.

Abstract

Lines 29-32 – the background highlights the role played by different indigenous foods in addressing several problems, including diseases. Still, the food products covered in the study could not be linked to the above issues in the discussion.

Line 33-35 – the aim mentions knowledge on the preparation and cooking of indigenous foods, but the study doesn't show any evidence of such a section being covered anywhere throughout the results.

Line 47- 50. The study's conclusion sounds too broad based on what was covered in the survey. I think the conclusion needs to align with the aim of the study, which will indicate if the surveyed communities possess the correct knowledge around the benefits of indigenous foods and the access or availability thereof. Now it's not clear if the potential is not realised because they don't consume the food or because they don't know its benefits.

Introduction

Line 64 – 113 correct sampling of district "sampling od districts,"

Sampling

Line 114 correct that would be used for the piloting to “that was used”

Sample size

Line 152 – be consistent, write ten percent in numbers and symbols

DISCUSSION

What makes the IFs better than the exotic foods, given that urban communities already buy exotic commodities, which could offer the same benefits? It would be better to promote the IFs so that the population can plant their own instead of just promoting that they buy the other in place of the other, unless the indigenous foods are cheap but overlooked due to a lack of knowledge.

Indigenous food availability

Since IFs play a significant role in reducing non-communicable diseases, were the assessed IFs part of the important ones or not? If not, then what does that mean about the population in the area, particularly as far as diseases are concerned?

What foods were planted in home gardens? There were no signs of IFs, and some gardens had IFs.

Limitations of the study

Maybe an explanation around the supermarkets used, if they were formal or informal supermarkets, such terminology would help one to have a clear picture of the establishments used for the study, or the difference thereof. This needs to be clarified under methodology.

The IFs used might have been influenced by access, since only starch was available, with no vegetables. This could speak to the supermarket's standards and storage conditions for certain food Items. Other IFs are highly perishable; hence, many food stalls or supermarkets shy away from purchasing them to avoid wastage.

Conclusion and recommendation

To overcome economic burden, people are bound to buy essential food products and never worry about their health benefits. Considering that the participants were at various income levels, was there any difference in their buying patterns, such as buying more health foods than their counterparts? This does not come up under discussion.

Maybe a comparison between the IFs and exotic foods in terms of which they buy or eat in place of what IFs will help us understand if buying such foods is influenced by knowledge or its purely accessibility. Also, comparing prices will help us understand why they went for expensive, exotic foods over cheap, healthy ones.

There's no link between the available IFs and their role in addressing malnutrition and non-communicable diseases, including their health benefits. Such a discussion will help us understand the health risks of the population.

Reviewer #2: Dear Author,

Thank you for submitting your work for consideration for publication in PLOS One. The manuscript is relevant and sound, however, corrections will be needed to improve its quality. Below are my comments and suggestions:

(1) Abstract, very sound

(2) Introduction, it is sound, however, authors must introduce IFs to the reader, what are they, nutritional composition, phytochemicals, examples of IFs in South Africa, and their contribution to the livelihood of households. Furthermore, previous studies done on IFs in South Africa should be included to show the knowledge gap and novelty of the current study. The objective of the study must be clear on line 81.

(3) Methods, line 84 is not clear since authors indicated that the aim is to develop a profile of the status of IFs amongst adults between 18 and 80. How are they going to develop the profile? The statement must be revised. Line 96, remove "of the Cape Metropole" since it is a repetition of the previous line. Line 112, change "od" to "of districts". Line 114, is it "would be" or "was"? The same comment is applicable for lines 129 and 133.

(4) The interpretation of the results is very sound.

(5) Discussion, under indigenous food availability, authors should expand the discussion and explain the reason(s) for butternut, maize meal, samp and beans, samp and white-fleshed sweet potato as the most widely available IFs. The same is applicable for green leafy vegetables not featuring with regards to availability in this study. Otherwise, the discussion is very sound with scientific tone.

(6) Limitation of the study and conclusion, they are very sound.

**Do you want your identity to be public for this peer review?** For information about this choice, including consent withdrawal, please see our Privacy Policy

Reviewer #1: No

Reviewer #2: No

---

## [Author Response · Author response to Decision Letter 1]

27 Aug 2025

Response: We have ensured that the manuscript is complaint.

Response: Funding details have been placed in appropriate sections.

Response: The University of Stellenbosch requires that the following is completed and approved by the Health Research Ethics Committee (HREC) before sharing data.

HREC MTA/DTA requirements, processes and term sheet:

• Material Transfer Agreement (MTA) and Data Transfer Agreement (DTA) HREC requirements and processes

• Data Transfer Agreeement (DTA): Guidance for Researchers

• HREC Termsheet: Material/Data Transfer Agreement

Data will be made available following the University policies upon request.

Response: Data requests may be sent to the corresponding author who is the data owner and will ensure compliance with the University policies for access as noted above. In addition, please note that in terms of the Protection of Personal Information Act No. 4 of 2013 of Republic of South Africa, Signed into law: 19 November 2013, published in the Government Gazette: 26 November 2013, Commencement dates: Full enforcement began on 1 July 2021b) there are also restrictions providing names of individuals of the Institutions other than committee details. Here is the contact details of HREC. http://www.sun.ac.za/english/faculty/healthsciences/rdsd/Pages/Health-Research-Ethics.aspx

b. If there are no restrictions, please upload the minimal anonymized data set necessary to replicate your study findings to a stable, public repository and provide us with the relevant URLs, DOIs, or accession numbers. For a list of recommended repositories, please see

Response: Not applicable.

4. We note you have included a table to which you do not refer in the text of your manuscript. Please ensure that you refer to Table 3 and 4 in your text; if accepted, production will need this reference to link the reader to the Table.

Responses: The comment has been implemented. It is actually Tables 2 and 3, and these have been referred to in the text and it is highlighted.

Response: The manuscript has been reviewed to ensure all citations are relevant.

---

## [Editor Report · Decision Letter 1]

8 Sep 2025

Dear Dr. Mbhenyane,

Thank you for submitting your manuscript to PLOS ONE. After careful consideration, we feel that it has merit but does not fully meet PLOS ONE’s publication criteria as it currently stands. Therefore, we invite you to submit a revised version of the manuscript that addresses the points raised during the review process.

The authors have satisfactorily addressed the reviewer comments. The data presentation is adequate but I am concerned it will be a dry read as there are no Figures to complement the Tables.

Under study design

- include coordinates of the "Cape Metropole". Some kind of map would also be useful

Under data analysis

- Delete "Quantitative data was captured into a Microsoft Excel spreadsheet and cleaned by the principal investigator."

- Specify the name, and version of the statistical software used.

We look forward to receiving your revised manuscript.

Kind regards,

Edwin Hlangwani, PhD

Academic Editor

PLOS ONE
---

## [Author Response · Author response to Decision Letter 2]

9 Sep 2025

Monday, 08 September 2025

Rebuttal letter

The responses are presented below each point .

Thank you for submitting your manuscript to PLOS ONE. After careful consideration, we feel that it has merit but does not fully meet PLOS ONE’s publication criteria as it currently stands. Therefore, we invite you to submit a revised version of the manuscript that addresses the points raised during the review process.

Response:

The authors have satisfactorily addressed the reviewer comments. The data presentation is adequate but I am concerned it will be a dry read as there are no Figures to complement the Tables.

Response:

Under study design

- include coordinates of the "Cape Metropole". Some kind of map would also be useful

Response:

Under data analysis

- Delete "Quantitative data was captured into a Microsoft Excel spreadsheet and cleaned by the principal investigator."

Response:

- Specify the name, and version of the statistical software used.

Response:

---

## [Editor Report · Decision Letter 2]

12 Sep 2025

Knowledge of Health Benefits, Availability, Accessibility and Consumption of Indigenous Foods by Urban Adults in the Cape Metropole, South Africa

PONE-D-24-41163R2

Dear Dr. Mbhenyane,

We’re pleased to inform you that your manuscript has been judged scientifically suitable for publication and will be formally accepted for publication once it meets all outstanding technical requirements.

Kind regards,

Edwin Hlangwani, PhD

Academic Editor

PLOS ONE
---

## [Editor Report · Acceptance letter]

PONE-D-24-41163R2

PLOS ONE

Dear Dr. Mbhenyane,

I'm pleased to inform you that your manuscript has been deemed suitable for publication in PLOS ONE. Congratulations! Your manuscript is now being handed over to our production team.

Kind regards,

on behalf of

Dr. Edwin Hlangwani

Academic Editor

PLOS ONE